# Frequency, clinical characteristics and outcomes of *Tropidolaemus* species bite envenomations in Malaysia

Ahmad Khaldun Ismail[1]*, Muhammad Nadzmi Hadi Abd Hamid[1], Nur Alissa Ariff[1], Vera Effa Rezar Frederic Ng[1], Wan Chee Goh[1], Nur Syafiqah Abdul Samat[1], Annuar Muhammad Zuljamal Osman[1], Ruth Sabrina Safferi[2], Zainalabidin Mohamed@Ismail[3]

1 Department of Emergency Medicine, Faculty of Medicine, Universiti Kebangsaan Malaysia, Kuala Lumpur, Malaysia, 2 Emergency and Trauma Department, Hospital Raja Permaisuri Bainun, Ipoh, Perak, Malaysia, 3 Emergency and Trauma Department, Hospital Tengku Ampuan Afzan, Kuantan, Pahang, Malaysia

* khaldun_ismail@yahoo.com

## Abstract

Pit vipers from the genus *Tropidolaemus* are identified as one of the common causes of snake bite from venomous species in Malaysia. All *Tropidolaemus* species bite cases referred to the Remote Envenomation Consultation Services (RECS) between 2015–2021 were included. A total of 4,718 snake-related injuries cases consulted to RECS with 310 (6.6%) involved *Tropidolaemus* species; of these 190 (61.3%) were *T. subannulatus* and 120 (38.7%) *T. wagleri*. All the *T. subannulatus* cases occurred in Sabah and Sarawak while all cases of *T. wagleri* occurred in Peninsular Malaysia. The majority of patients were male (74.8%) and adults between 18–59 years old (61.2%). The upper limb (56.6%) was the most frequent anatomical region involved. Most cases were non-occupationally related (75.4%). Bites from both species caused local pain (77.1%) and swelling (27.2%). Most patients complained of mild pain (48.0%). Paracetamol (40.0%) was the most common analgesic prescribed. Antivenom was not indicated in all cases. Two patients were given antivenom inappropriately before RECS consultation. Most patients (54.7%) needed hospital observation for less than 24 hours. No deaths occurred in the group studied.

## Author summary

There is poor documentation and awareness of the clinical characteristics and significance of *Tropidolaemus* species bite in Malaysia. This study analysed RECS consultation data from 2015–2021 regarding *Tropidolaemus* species bite cases in Malaysia. By analysing the RECS consultation log, the frequency, geographical distribution, clinical features, clinical management and outcomes of verified *Tropidolaemus* species bite were documented. The significance of this study is to provide verified and reliable information regarding the clinical characteristics, significance and clinical burden of *Tropidolaemus* species bite cases. We provide the evidence for the revision and exclusion of *T. subannulatus* from the WHO category 2 list of medically important venomous snakes in Malaysia. By identifying

**Data Availability Statement:** All relevant data are within the manuscript and its Supporting Information files.

**Funding:** This work is supported by the Universiti Kebangsaan Malaysia, Faculty of Medicine, Fundamental Grant [FF-2022-024] to AKI (https://www.ukm.my/fper/).The funders had no role in study design, data collection and analysis, decision to publish, or preparation of the manuscript.

**Competing interests:** The authors have declared that no competing interests exist.

and understanding the characteristics and significance of *Tropidolaemus* species bite envenomation, optimal management can be provided for a favourable outcome.

## Introduction

Snake related injury (SRI) is a potentially fatal illness produced either by mechanical injury or envenomation from a snake. Envenomation can occur during a venomous snake bite and may cause topical injury when venom is sprayed into the eyes by those snakes that have the ability to spray venom as a defence mechanism [1]. According to the WHO, 4.5–5.4 million individuals are bitten by snakes annually with 1.8–2.7 million experiencing significant clinical effects, and 81,000–138,000 deaths as a result of complications. Many sufferers do not seek treatment at health centres or hospitals, preferring to rely on traditional methods or due to inadequacy of health services. There are limited epidemiological data in Malaysia due to under recognition of snake related injuries as a notifiable disease. A total of 15,798 snake bite frequencies were reported in Malaysia between 2010 and 2014. During the same period, there were 16 deaths, averaging 3 to 4 fatalities per year [2]. The states with the most snake bite frequencies are Kedah and Perak probably because of agricultural activities. Unfortunately, there is no detailed data on the species of snakes involved, the type of injury, geographical distribution, clinical features, clinical management and outcome [3].

There are more than 200 species of snakes in Malaysia with approximately 20% being medically significant [3]. Among the pit vipers in Malaysia, the *Tropidolaemus* species is one of the common causes for venomous snake bites [4]. There are two subspecies of *Tropidolaemus* in Malaysia, *T. wagleri* and *T. subannulatus*. *T. wagleri* is indigenous to Peninsular Malaysia, *T. subannulatus* is indigenous to Borneo [5,6]. *Tropidolaemus* species are arboreal and generally found in wet lowland rainforest from sea level up to 400m. Both species feed on small animals especially birds, frogs, lizards, mice and other rodents [5–8].

There are significant ontogenetic morphological differences between male and female *Tropidolaemus* snakes [5–8]. Adult males can grow up to a length of 52 cm and females between 92–96 cm. Juvenile *Tropidolaemus* typically have a green, slender body with white and red spots. As they mature, males and females diverge in patterning and colouration in both species (Fig 1). Adult male *Tropidolaemus* species remain almost the same as juveniles with a white and red postocular stripe. Adult female *T. wagleri* morphology evolves with the development of black and yellow crossbars on top of the body, a black postocular stripe and a banded belly. Adult female *T. subannulatus* have greenish-blue body with turquoise crossbars and a cream or yellow postocular stripe.

All pit vipers are venomous. However, *T. wagleri* and *T. subannulatus* are generally not considered to be aggressive and the envenomation is atypical among many pit vipers. Venom from *T. wagleri* is only weakly pseudo-procoagulant, clotting fibrinogen with only a negligible net anticoagulant effect. *T. subannulatus* venom does not affect clotting [9]. The venom proteome of *T. wagleri* has abundant neurotoxic peptides (Waglerins) causing neurotoxic envenomation in mice [10–12]. For both species bite envenomation in humans is reported to be limited to local effects such as pain, mild oedema, erythema and paraesthesia [3]. No specific antivenom has been developed for these two species as it did not meet the WHO's criteria for antivenom production [13]. Malaysia do not manufacture antivenom, however due to the closely related medically significant venomous land snake species to Thailand, the country resort to import the appropriate antivenom from Queen Saovabha Memorial Institute (Thai Red Cross). [1–3, 6, 10]

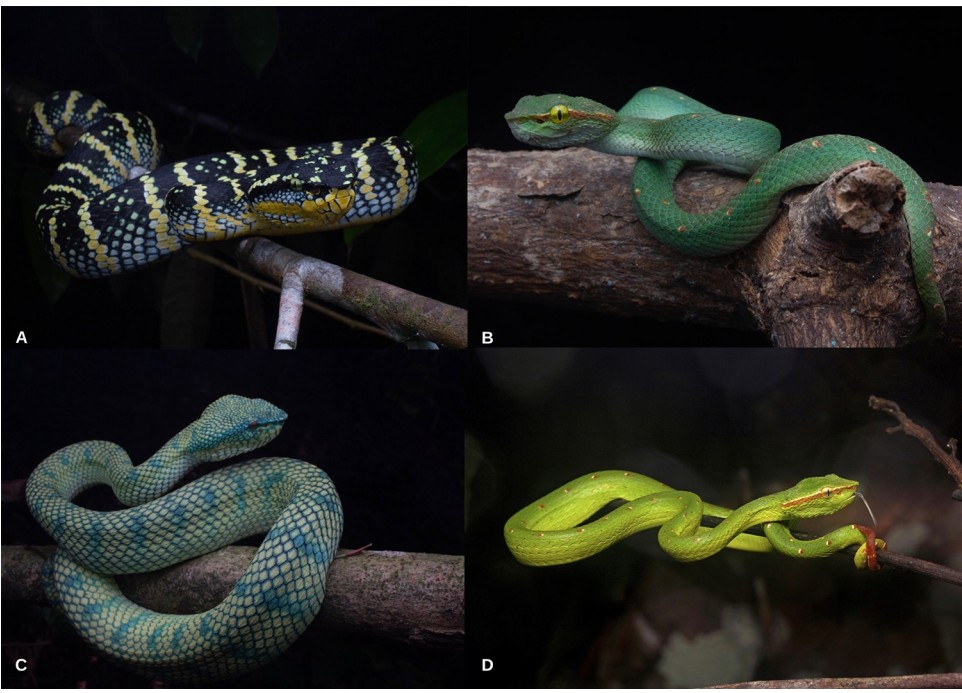

**Fig 1. The morphological differences between male and female *Tropidolaemus wagleri* and *Tropidolaemus subannulatus*.** (A) Adult female *Tropidolaemus wagleri*. *Image credit*: *M. K. Arif* (B) Adult male *Tropidolaemus wagleri*. *Image credit*: *M. K. Arif* (C) Adult female *Tropidolaemus subannulatus*. *Image credit*: *M. K. Arif* (D) Adult male *Tropidolaemus subannulatus*. *Image credit*: *E. W. Teo*.

The Remote Envenomation Consultancy Services (RECS) is a specialised risk management support system [1–3]. Since its inception in 2012, RECS has assisted clinicians with 24-hour volunteer 'on-call' consulting service for clinical management for snake bite, and bites or stings from other animals. RECS Consultants are Emergency Physicians with a special interest in clinical toxinology who are members of the Malaysian Society of Toxinology (MST). Through RECS consultation, clinical data on each patient's management is documented. Requests for consultations are received from various specialties such as emergency medicine, internal medicine, paediatrics, anaesthetics, orthopaedics and pharmacists. This provides a unique opportunity for direct communication with various specialties managing the same case, ensuring a continuum of appropriate care and documentation. RECS also provides the opportunity to minimise medicolegal issue among healthcare professionals and optimise patient care through direct communication with experts in the field and allows active teaching and learning to take place during the consultation.

The objectives of this study were to determine the frequency, geographical distributions, clinical features, clinical management and outcomes of *Tropidolaemus* species bite envenomation in Malaysia consulted to Remote Envenomation Consultation Services (RECS) from 2015 to 2021. This study provides verified and reliable information regarding the clinical characteristics, significance and clinical outcomes of *Tropidolaemus* bite cases.

## Methods

This is a retrospective cohort study of bites from *Tropidolaemus* species in Malaysia referred to RECS from 2015 to 2021. The study was approved by the institutional Research and Ethics committee Faculty of Medicine, National University of Malaysia prior to data collection. All

data collected from RECS database were kept anonymous and confidential. A universal sampling method was used whereby all case details were collected from RECS consultation log and case record. Only confirmed *Tropidolaemus* species bite cases were included. The species confirmation was verified by RECS experts during each consultation based on the specimen brought to the hospital or the picture of the actual specimen taken. The data for each year was documented separately in the data collection sheet and descriptively analysed.

## Results

From 2015 to 2021, there were a total of 5,820 consultations to RECS. 4,718 (81.0%) were snake related injury (SRI) cases (Fig 2). Annual consultation varied from 345 in 2015 to 880, in 2018. The number of identified SRI was 2,165 with 310 confirmed *Tropidolaemus* spp. Of these, 190 (61.3%) were *T. subannulatus* and 120 (38.7%) *T. wagleri* cases.

The majority of cases were male (n = 232, 74.8%) and between 18 to 59 years old (n = 190, 61.3%) (Table 1). Most involved Malaysians (n = 263, 84.8%) and were non-occupational related (n = 233, 75.2%). The majority of the cases happened during the daytime (n = 261, 84.2%) and outdoors (n = 274, 88.4%).

The majority of consults for *T. subannulatus* cases were from Sarawak (n = 150, 78.9%) and the majority of consults for *T. wagleri* were from Perak (n = 63, 52.5%) (Fig 3 and S1 Table).

The majority of cases were bitten once (n = 274, 88.4%), followed by twice (n = 32, 10.3%) and more than two times (n = 2, 0.6%) and undocumented (n = 2, 0.6%). The most common anatomical region affected was the upper limb (n = 184, 56.6%), followed by lower limb (n = 123, 37.8%) (Fig 4).

There were 163 (52.6%) cases performed inappropriate first aid intervention prior to hospital arrival. Tourniquet is the commonest (n = 97, 59.5%). The most common symptoms and signs were local pain at the site of bite (n = 256, 76.9%) and swelling (n = 154, 27.3%) (Table 2). The majority of cases documented mild pain (n = 149, 48.1%) at presentation. The most frequent analgesia administered was Paracetamol (n = 132, 40.2%), followed by Tramadol (n = 56, 17.1%).

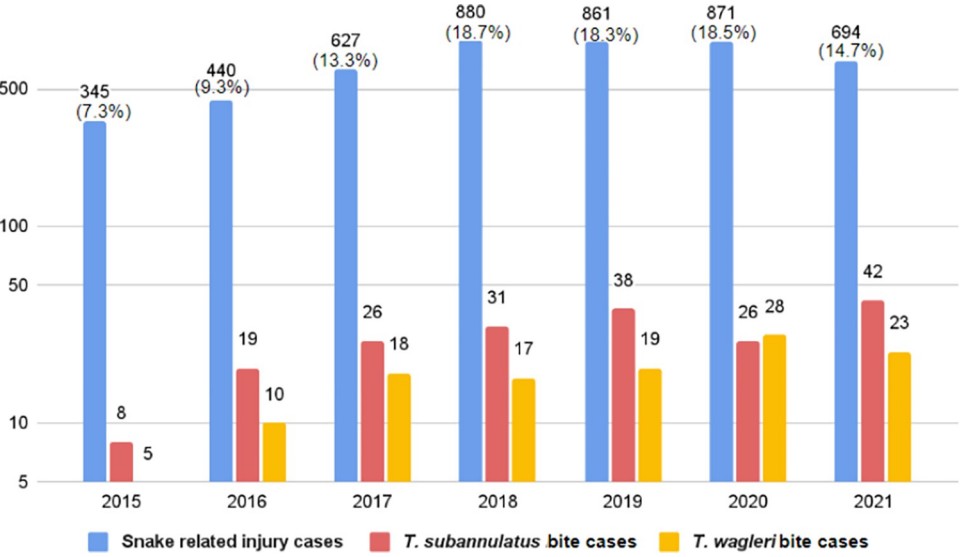

**Fig 2. Annual number of snake related injury consulted to RECS and the number of *Tropidolaemus subannulatus* and *Tropidolaemus wagleri* bite cases from 2015 to 2021.**

**Table 1. Demographic characteristics of the *Tropidolaemus* spp. snake bite patients.**

| Demographic characteristics | | Snake bite victims n (%) |
|---|---|---|
| Gender | Male | 232 (74.8) |
| | Female | 77 (24.8) |
| | Undocumented | 1 (0.3) |
| Age | Below 7 years old | 20 (6.5) |
| | 7–12 years old | 21 (6.8) |
| | 13–17 years old | 14 (4.5) |
| | 18–39 years old | 90 (29.0) |
| | 40–59 years old | 100 (32.3) |
| | 60 years old and above | 63 (20.3) |
| | Undocumented | 2 (0.6) |
| Nationality | Citizen | 263 (84.8) |
| | Non-citizen | 22 (7.1) |
| | Undocumented | 25 (8.1) |
| Occupational-related injury | Yes | 73 (23.5) |
| | No | 233 (75.2) |
| | Undocumented | 4 (1.3) |
| Time of incident | 00:00–06:59 | 15 (4.8) |
| | 07:00–11:59 | 126 (40.6) |
| | 12:00–18:59 | 135 (43.5) |
| | 19:00–23:59 | 32 (10.3) |
| | Undocumented | 2 (0.6) |
| Location of incident | Indoor | 30 (9.7) |
| | Outdoor | 274 (88.4) |
| | Undocumented | 6 (1.9) |

Antivenom was not indicated for either *Tropidolaemus* spp. bites in this study. However, there were two cases where inappropriate antivenom was administered without any complication. Intravenous antibiotic, Cloxacillin was administered to two patients, however the indication for this was undocumented. There was no surgical intervention performed.

The majority of patients were observed in hospital for less than 48 hours (n = 223, 71.9%) (Table 2). However, in 79 (25.5%) the duration of hospital stay was undocumented. No mortality or longer-term morbidity was reported.

## Discussion

*Tropidolaemus* spp. were the most commonly identified SRI cases referred to RECS between 2015–2021. The trend of frequency of *Tropidolaemus* bite cases has been increasing throughout the 7 years. Many factors lead to the high frequency of *Tropidolaemus* bite cases including human activities, destruction of natural habitat, climate change or natural disaster. The rapid development of residential, recreational and industrial areas encroaching into snake habitat is likely to be the major factor [3]. *T. wagleri* gets its common name, Temple pit viper from The Snake Temple in Penang, Malaysia. This was built to honour Chor Soo Kong, a Buddhist monk from the Song Dynasty, for his deeds in healing the sick and giving shelter to snakes. After the temple was built, snakes started taking shelter in the temple, particularly *T. wagleri* [14].

This is the first study of the frequency, clinical characteristics and outcomes of *Tropidolaemus* spp. bite envenomation in Malaysia. The majority of patients were adult Malaysian male. This may be due to their greater exposure to outdoor activities and suggest that patients were encroaching into the snake's natural habitat such as forests, oil palm plantations and farms rather than the snakes venturing into human habitats. This correlates with the natural habitat of *Tropidolaemus* spp. which is wet lowland forest area. Most cases occurred in daytime.

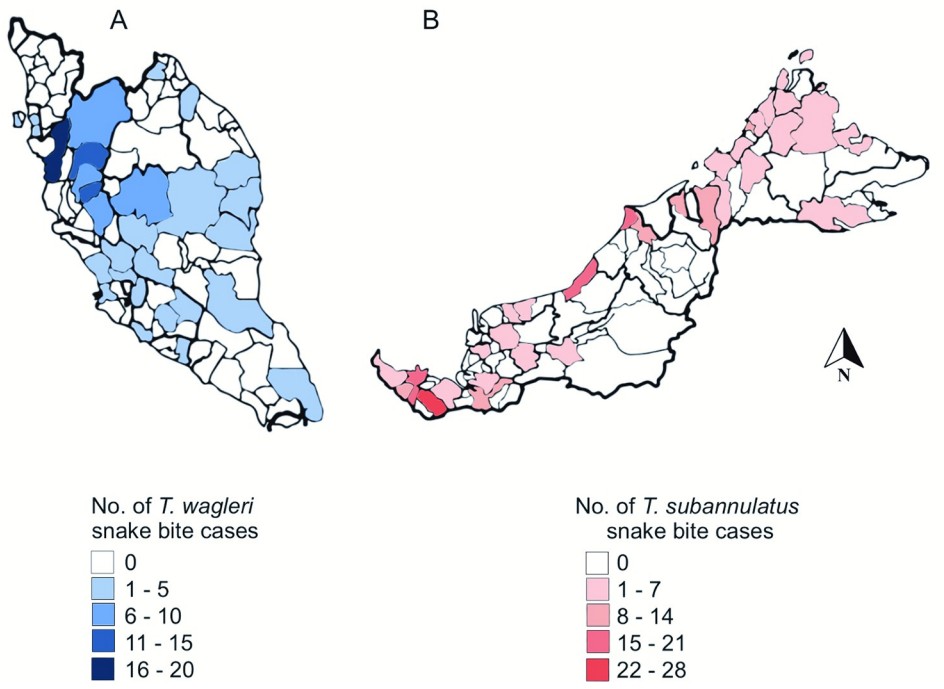

**Fig 3. Number of *Tropidolaemus wagleri* and *Tropidolaemus subannulatus* cases consulted to RECS from each state's district from 2015–2021.** (A) Geographical distribution of *T. wagleri* in Peninsular Malaysia. The highest consultations was recorded in the district of Larut Matang, Perak. (B) Geographical distribution of *T. subannulatus* in East Malaysia (Sabah and Sarawak). The highest consultations was recorded in the district of Serian, Sarawak. Base map and data from OpenStreetMap and OpenStreetMap Foundation.(© OpenStreetMap contributors, https://www.openstreetmap.org/#map=6/4.226/108.237).

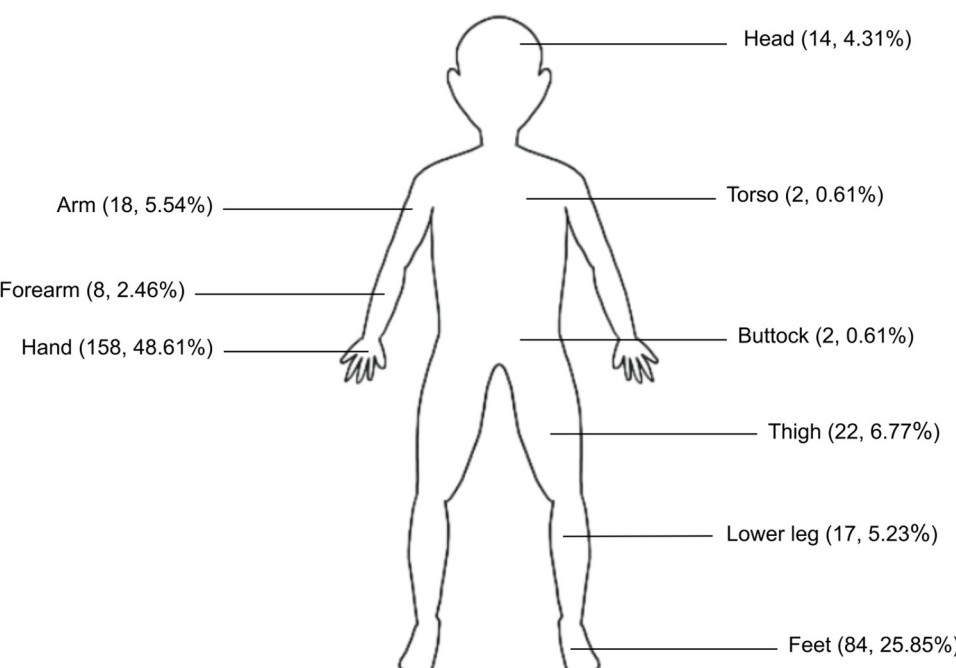

**Fig 4. The frequency of anatomical region involved in bite injury from *Tropidolaemus* spp.**

**Table 2. The clinical presentation, management and outcome of *Tropidolaemus spp*. bite consulted to RECS from 2015–2021.**

| | | | n (%) |
|---|---|---|---|
| First aid intervention | No | Tourniquet | 147 (47.4%) |
| | Yes | | 163 (52.6%) |
| | | | 97 (59.5%) |
| | | Ointment | 37 (22.7%) |
| | | Squeezed | 41 (25.2%) |
| | | Sucked | 8 (4.9%) |
| | | Washed | 27 (16.6%) |
| | | Others (coconut oil, vinegar, massage, salt, lime juice, eat snake tail, leaves) | 9 (5.5%) |
| Symptoms | Local pain | | 256 (76.9%) |
| | Numbness | | 22 (6.6%) |
| | Asymptomatic | | 43 (12.9%) |
| | Others | | 8 (2.4%) |
| | Undocumented | | 4 (1.2%) |
| Signs | Bite site | | 265 (47.0%) |
| | Swelling | | 154 (27.3%) |
| | Bleeding | | 59 (10.5%) |
| | Erythematous | | 40 (7.1%) |
| | Tender | | 19 (3.4%) |
| | Bruising | | 3 (0.5%) |
| | No sign | | 5 (0.9%) |
| | Others | | 17 (3.0%) |
| | Undocumented | | 2 (0.4%) |
| Pain Score | None | | 54 (17.4%) |
| | Mild (1–3) | | 149 (48.1%) |
| | Moderate (4–6) | | 69 (22.3%) |
| | Severe (7–10) | | 14 (4.5%) |
| | Undocumented | | 24 (7.7%) |
| Analgesics | Paracetamol | | 132 (40.2%) |
| | Tramadol | | 56 (17.1%) |
| | Diclofenac | | 16 (4.9%) |
| | Fentanyl | | 2 (0.6%) |
| | Morphine | | 1 (0.3%) |
| | Mefenamic acid | | 1 (0.3%) |
| | Nil | | 114 (34.8%) |
| | Undocumented | | 6 (1.8%) |
| Length of hospital stay | < 24 hours | | 170 (54.8%) |
| | 24–48 hours | | 53 (17.1%) |
| | 48–72 hours | | 8 (2.6%) |
| | Undocumented | | 79 (25.5%) |

Although *Tropidolaemus* spp. are considered nocturnal species, human activities during daytime may have contributed to human-snake conflicts.

In this study, the majority of the cases involved patients being bitten only once. This is probably due to the snake's defensive nature. *Tropidolaemus* species only strike to bite when provoked, disturbed or accidentally stepped on. Most of the cases involved the patients being bitten while plucking fruits in fruit orchards, gardening, weeding, farming and walking in the dark. In a few cases patients were bitten more than once. The majority were bitten on the

upper limb, especially the hands. This is because *Tropidolaemus* spp. are arboreal and usually found above the ground. However, a significant number of patients were bitten on the lower limb, especially on the foot. The mechanism of injury for these patients was from accidentally stepping on the snake.

Local pain, puncture wounds, localised swelling, bleeding, and erythema were the most common effects of *Tropidolaemus* spp. bite. Vasoactive substances and enzymatic toxins interact to cause localised swelling and erythema. *Tropidolaemus* spp. envenomation has not been shown to induce shock and coagulopathy. Our study shows that the majority of bites only result in mild-to-moderate local envenomation presenting with pain and swelling. No patients developed tissue necrosis, cardiovascular instability or coagulopathy.

There appears to be still high frequency of inappropriate first aid performed by snake bite patients in Malaysia. This practice may have been widely practiced due to poor awareness among the public. There are limited published articles related to prehospital intervention for snakebite in Malaysia. This is partly because of lack of data and understanding of the situation and practice in Malaysia. The socioeconomic status and health services in Malaysia may differ from other countries, therefore the prehospital intervention may also differ in terms of types of interventions and health seeking behaviour of the community. The availability and accessibility of healthcare services has a direct impact on prehospital practices and intervention. A better awareness, knowledge and understanding about this issue may help to reduce or minimize complication from inappropriate prehospital intervention. Inappropriate intervention can be mitigated by targeted community awareness programs such as first aid workshop, seminar and disseminating appropriate information through various electronic and printed media.

Prompt management is important in any snake bite cases. We found that the type of analgesics used positively correlated with the patient's pain score. Since most patients experienced only mild pain, analgesics such as Paracetamol were given. Antibiotics in snake bite cases may be indicated for patients developing secondary bacterial infection. In this study, two patients were given intravenous antibiotics, however the indications were undocumented.

Almost all patients sustained only minimal local envenoming. They did not progress to serious local or systemic envenomation that requires antivenom or special treatment. However, two patients in this study were given antivenom inappropriately. The attending physicians had not obtained expert consult prior to giving antivenom. In the WHO category listings of medically important venomous snakes, *T. subannulatus* is categorised as high priority for antivenom administration [15], *T. subannulatus* is listed under the category of venomous snakes that are capable of causing morbidity, disability and mortality, in which the exact clinical data are still lacking. To date, there is no specific antivenom manufactured for *Tropidolaemus* species. However, our study suggests that the clinical effects of bite envenomation from *Tropidolaemus* spp. are mild and that the WHO Category 2 listing should be revised.

## Study limitations

This study is limited to confirmed cases of *Tropidolaemus* spp. bite in Malaysia. There may be more *Tropidolaemus* spp. cases classified as unidentified snake bite, however it provides good representation of the frequency and geographical distribution in the country. Data was collected from cases that were referred to RECS and do not represent all snake-related injury cases that may have occurred in Malaysia during the study period. There is no data on patients' progress during follow-up or psychological effects, however, all patients were discharged well. Further studies on venom components and toxicity of *Tropidolaemus* spp on human are recommended. Research on snake bite identification kit, especially for unidentified snake bite cases would be beneficial in the future.

## Conclusion

This study has made us understood better the clinical characteristics, outcomes and appropriate clinical management of *Tropidolaemus subannulatus* and *Tropidolaemus wagleri* bites in Malaysia. Public awareness regarding snake bite is important to reduce the frequency of snake bite and its outcomes. Attending physicians should have adequate knowledge and skills to manage these patients. Proper identification of the snake species and appropriate diagnosis leads to the optimal management for patients. *Tropidolaemus* spp. bite envenomation caused mild local symptoms. We suggest that human envenomation from *Tropidolaemus* spp. does not require antivenom and *Tropidolaemus subannulatus* should be excluded from the WHO Category 2 list of medically important venomous snakes in Malaysia. Symptomatic clinical management with analgesia and short period of close serial monitoring are sufficient to manage these patients.

## Supporting information

**S1 Table. The number of *Tropidolaemus* spp. bite cases consulted to RECS from each district and state in Malaysia from 2015–2021.**
(PDF)

## Acknowledgments

We are grateful to all consultants from Remote Envenomation Consultation Services (RECS) for their tremendous efforts in improving the quality of patients care in Malaysia. We acknowledge Puan Nur Hazwanie Abd Halim and Puan Nurfarhana Hizan Binti Hijas from Malaysian Biodiversity Information System (MyBIS) and Puan Nurul Saadah Binti Ahmad from Department of Emergency Medicine, UKM for the technical support. We thank Professor Colin Robertson from University of Edinburgh for reviewing the manuscript.

## Author Contributions

**Conceptualization:** Ahmad Khaldun Ismail.

**Data curation:** Ahmad Khaldun Ismail, Muhammad Nadzmi Hadi Abd Hamid, Nur Alissa Ariff, Vera Effa Rezar Frederic Ng, Wan Chee Goh, Nur Syafiqah Abdul Samat.

**Formal analysis:** Ahmad Khaldun Ismail, Muhammad Nadzmi Hadi Abd Hamid, Nur Alissa Ariff, Vera Effa Rezar Frederic Ng, Wan Chee Goh, Nur Syafiqah Abdul Samat.

**Funding acquisition:** Ahmad Khaldun Ismail.

**Investigation:** Ahmad Khaldun Ismail, Muhammad Nadzmi Hadi Abd Hamid, Nur Alissa Ariff, Vera Effa Rezar Frederic Ng, Wan Chee Goh, Nur Syafiqah Abdul Samat.

**Methodology:** Ahmad Khaldun Ismail, Muhammad Nadzmi Hadi Abd Hamid, Nur Alissa Ariff, Vera Effa Rezar Frederic Ng, Wan Chee Goh, Nur Syafiqah Abdul Samat.

**Project administration:** Ahmad Khaldun Ismail.

**Resources:** Ahmad Khaldun Ismail.

**Supervision:** Ahmad Khaldun Ismail.

**Validation:** Ahmad Khaldun Ismail.

**Visualization:** Ahmad Khaldun Ismail, Muhammad Nadzmi Hadi Abd Hamid, Nur Alissa Ariff, Vera Effa Rezar Frederic Ng, Wan Chee Goh, Nur Syafiqah Abdul Samat.

**Writing – original draft:** Ahmad Khaldun Ismail, Muhammad Nadzmi Hadi Abd Hamid, Nur Alissa Ariff, Vera Effa Rezar Frederic Ng, Wan Chee Goh, Nur Syafiqah Abdul Samat.

**Writing – review & editing:** Ahmad Khaldun Ismail, Muhammad Nadzmi Hadi Abd Hamid, Nur Alissa Ariff, Vera Effa Rezar Frederic Ng, Wan Chee Goh, Nur Syafiqah Abdul Samat, Annuar Muhammad Zuljamal Osman, Ruth Sabrina Safferi, Zainalabidin Mohamed@Ismail.

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
