## [Decision Letter · Decision Letter 0]

24 Oct 2022

Dear Dr Ismail,

Thank you very much for submitting your manuscript "Incidence, clinical characteristics and outcomes of Tropidolaemus species bite envenomation in Malaysia" for consideration at PLOS Neglected Tropical Diseases. As with all papers reviewed by the journal, your manuscript was reviewed by members of the editorial board and by several independent reviewers. In light of the reviews (below this email), we would like to invite the resubmission of a significantly-revised version that takes into account the reviewers' comments. 

We cannot make any decision about publication until we have seen the revised manuscript and your response to the reviewers' comments. Your revised manuscript is also likely to be sent to reviewers for further evaluation.

Sincerely,

Wayne Hodgson

Academic Editor

José María Gutiérrez

Section Editor

Reviewer's Responses to Questions

**Key Review Criteria Required for Acceptance?**

**Methods**

-Are the objectives of the study clearly articulated with a clear testable hypothesis stated?

-Is the study design appropriate to address the stated objectives?

-Is the population clearly described and appropriate for the hypothesis being tested?

-Is the sample size sufficient to ensure adequate power to address the hypothesis being tested?

-Were correct statistical analysis used to support conclusions?

-Are there concerns about ethical or regulatory requirements being met?

Reviewer #1: The manuscript describes the incidence of envenomation by Tropidolaemus snakes in Malaysia, during the period of 2015 to 2021. Signs and symptoms, as well as the outcomes of these envenomation are also described. The authors clearly pointed out the aim of the study and describe the affected population in appropriated tables and figures.

Reviewer #2: -Are the objectives of the study clearly articulated with a clear testable hypothesis stated? 

Yes

-Is the study design appropriate to address the stated objectives?

yes

-Is the population clearly described and appropriate for the hypothesis being tested?

yes

-Is the sample size sufficient to ensure adequate power to address the hypothesis being tested?

Yes

-Were correct statistical analysis used to support conclusions?

NA

-Are there concerns about ethical or regulatory requirements being met?

They need to clarify how did they identify the species of snakes.

Reviewer #3: The objectives of the study were not clearly articulated with the hypothesis stated, as identification of the Tropidolaemus species was not clearly detailed. This aspect is a key point and should be completed. As there are more than 200 species of snakes in Malaysia, is there any possibility of no venomous snakes to mimic the behavior or the appearance of Tropidolaemus species snakes, including the ontogenetic variability in the morphology? This would be important to understand in which extend such variations could contribute to a misdiagnosed snakebite envenoming or not. 

Another limitation of the study is related to the dataset, composed of cases reported remotely. In this situation, it should be clearly stated how authors ensured precise identification of the causative agent of the injuries.

**Results**

-Does the analysis presented match the analysis plan?

-Are the results clearly and completely presented?

-Are the figures (Tables, Images) of sufficient quality for clarity?

Reviewer #1: The manuscript presents the results in an appropriate number of tables and figures.

Reviewer #2: -Does the analysis presented match the analysis plan?

-Are the results clearly and completely presented?

Yes, but they need to better discuss their findings. 

-Are the figures (Tables, Images) of sufficient quality for clarity?

yes, but can be improved.

Reviewer #3: In general, snakebite envenoming is considered an occupational event during agricultural activities. In the present study, however, most cases were referred as non-occupational injuries. Could authors describe in which circumstances the cases occurred? Were these conditions compatible with Tropideolaemus behavior characteristics?

A full map of the country would be more illustrative of the geographical distibution of the two Tropidolaemus species. A more exhaustive discuss of the dicotomic distribution of T. wagleri and T. subannulatus in the two Malaysian states is missing.

As there is no specific antivenom for Thropidolaemus envenoming available, which antivenom(s) is/are available in Malaysia for snakebite envenoming? Give the name of the manufacturer(s), characteristics of the product (especially neutralizing potency), doses indication, etc. Is there any study to assess the cross neutralization between antivenom(s) and T. subannulatus or T. wagleri snake venoms to justify the use in the country? 

Although sometimes expressed as the number of cases during a given period of time, by definition, incidence is a measure of the frequency or probability of occurrence of a given medical condition in a population within a specified period of time. No incidence rates were herein calculated, thus the term should be avoided, either in the text or table. Instead of incidence, prefer frequency.

Outcomes were not clearly described for those who stayed at the hospital more than 24hrs (61 cases), as it does not correspond to those with severe (14) and moderate (69) pain. Were there refractory cases to analgesics?

**Conclusions**

-Are the conclusions supported by the data presented?

-Are the limitations of analysis clearly described?

-Do the authors discuss how these data can be helpful to advance our understanding of the topic under study?

-Is public health relevance addressed?

Reviewer #1: The conclusion is generic and not directly related to the results presented.

Reviewer #2: -Are the conclusions supported by the data presented?

Yes, 

-Are the limitations of analysis clearly described?

Limitations are not fully in line with the aim of this work

-Do the authors discuss how these data can be helpful to advance our understanding of the topic under study?

Not quite.

-Is public health relevance addressed

Somehow, authors should give more emphasis bout this.

Reviewer #3: Conclusions are not strongly supported by the data presented as case definition of Tropidolaemus snakebite envenoming is not clearly stated. The remote identification of the two species constitutes a major limitation of the analysis. Authors suggest a revision in the WHO Category listing, which is premature, considering the limitation in the methodology of the study.

**Editorial and Data Presentation Modifications?**

Reviewer #1: (No Response)

Reviewer #2: Some paragraphs should be rephrased in order to improve readability and clarity.

Reviewer #3: (No Response)

**Summary and General Comments**

Reviewer #1: The manuscript is of importance, despite describing accidents caused by a snake with a low incidence of envenoming between snakebites in Malaysia, and with less severe intensity.

Reviewer #2: Summary and general comments. 

The manuscript ” Incidence, clinical characteristics and outcomes of Tropidolaemus species bite envenomation in Malaysia” (PNTD-D-22-01125) compiles the number of Tropidolaemus bites that were referred to the Remote Envenomation Consultation Services (RECS) for a period of six years. According to them, although T. subannulatus and T. wagleri are associated with high morbidity in the area, the use of antivenom to treat these snakebites is not prescribed. In addition, authors provide additional information ranging from the morphological characteristics of these snakes, seasons with the highest morbidity, demographics of patients, site of the bites and the typical first aid performed, e.g. 

This manuscript can be enhanced if authors review, analyze and discuss additional data regarding the venom from Tropidolaemus , its biochemistry, pharmacology and the way is neutralized. Is there any specific or para-specific antivenom? If there were a person with a high degree of envenomation what antivenom should be administer? 

Particular comments,

Author Summary

Line 45-47.- What is the purpose of excluding T. subannulatus from WHO category 2? Is this highly venomous or associated to high mortality? Please explain

Introduction.

Line 52.- How mechanical injury can be fatal? Please, explain or amend.

Line 52-55- How envenomation happens when venom is sprayed into the eyes? Please, explain or amend.

Line 59-60 .- Lack of cohesivity between these two paragraphs.

Line 60-62.- Authors are saying that there is a limited epidemiological data, but they provide exact numbers in previous and further paragraphs. Please, clarify.

In the text:” During the same period, there were 16 deaths averaging 3 to 4 fatalities per year” what species and what is the source of information? Please cite. 

First and second paragraph could be merged to make one cohesive and concise text. 

Line 73-75. “There are two subspecies of Tropidolaemus in Malaysia, T. wagleri and T. subannulatus. T. wagleri is indigenous to Peninsular Malaysia, WHEREAS T. subannulatus is indigenous to Borneo”. In here you have a distribution map, this could also be referred as Fig 1 or 2.

Line 93-103. Behavior and distribution of this snake are important. However, authors should focused more on providing as much information as available about venom composition, its biochemistry and antivenoms, rather than describing physical characteristics of these species. 

Line 106-118. In the last paragraph authors provide information that does not match with the aim of the work. Please re-write and make it sound with the objectives, findings and conclusions of it. 

Methods.

Line 106-118. What is the piece of evidence confirming that the envenomation was done by Tropidolaemus species and not by any other ? In this regard, how patients or physicians identified at the genus and species level? Please clarify and state that in the corresponding section. 

Results

 Line 136- 140. Again, how did they confirm the species?

Line 159. Figure 3. Not many people know Malaysian geography, it would be great to indicate as A and B the name of these two regions and write down the name (in the figure or figure description) of the districts where you found the highest incidence of these two species. 

Line 172-173.- site of the bite instead of “puncture wound”

Line 78.- Table 2 description is not clear. Please re-phrase it. 

Line 185.- what is Cloxacillin? Indicate. If this was undocumented how did you get this information?

Line 187-188.- Again, how undocumented information became available for authors?

 “T. wagleri gets its common name, Temple pit viper from The Snake Temple in Penang, Malaysia. This was built to honour Chor Soo Kong, a Buddhist monk from the Song Dynasty, for his deeds in healing the sick and giving shelter to snakes. After the temple was built, snakes started taking shelter in the temple, particularly T. wagleri ” This information is interesting. Nonetheless, authors should focus on more relevant data like venom composition, antivenoms, WHO information, public health management or even awareness or educational programs.

Line 224-225. This is a good opportunity to talk about venom composition. 

Line 232-233 This is a good opportunity to talk about specific antivenoms, availability, specificity, Etc.

Conclusion

-Authors mention “ Public awareness regarding snake bite is important to reduce the incidence of snake bite and its outcomes” . Please discussed why in the corresponding section.

-“We suggest that human envenomation from Tropidolaemus spp. does not require antivenom under normal circumstances” what is a normal circumstance? What kind of antivenom is recommended? 

- “Symptomatic clinical management with analgesia and short period of close serial monitoring are sufficient to manage these patients.” Why do you recommend this? Based on what study? Did authors measure the lethality of this venoms? Does it contain non-dangerous toxins?

References,

Please, Italicized scientific names and cite according to journals criteria.

Reviewer #3: (No Response)

PLOS authors have the option to publish the peer review history of their article (what does this mean?). If published, this will include your full peer review and any attached files.

Reviewer #1: No

Reviewer #2: No

Reviewer #3: No
---

## [Decision Letter · Decision Letter 1]

23 Nov 2022

Dear Dr Ismail,

We are pleased to inform you that your manuscript 'Frequency, clinical characteristics and outcomes of Tropidolaemus species bite envenomation in Malaysia' has been provisionally accepted for publication in PLOS Neglected Tropical Diseases.

Best regards,

Wayne Hodgson

Academic Editor

José María Gutiérrez

Section Editor

Reviewer's Responses to Questions

**Key Review Criteria Required for Acceptance?**

**Methods**

-Are the objectives of the study clearly articulated with a clear testable hypothesis stated?

-Is the study design appropriate to address the stated objectives?

-Is the population clearly described and appropriate for the hypothesis being tested?

-Is the sample size sufficient to ensure adequate power to address the hypothesis being tested?

-Were correct statistical analysis used to support conclusions?

-Are there concerns about ethical or regulatory requirements being met?

Reviewer #2: -Are the objectives of the study clearly articulated with a clear testable hypothesis stated?

Yes

-Is the study design appropriate to address the stated objectives?

YES

-Is the population clearly described and appropriate for the hypothesis being tested?

Yes

-Is the sample size sufficient to ensure adequate power to address the hypothesis being tested?

Yes

-Were correct statistical analysis used to support conclusions?

Yes

-Are there concerns about ethical or regulatory requirements being met?

No

Reviewer #3: All reviewers pointed out that it would be necessary to clarify how the identification of the species was perfomed. The diagnosis based on the snake brought to the hospital for identification or the pictures of the animal that caused the bite does not guarantee that there were the only species of Tropidolaemus snakes involved. Which was the proportion of snakes verified by the experts? Were these experts graduated herpetologists?

**Results**

-Does the analysis presented match the analysis plan?

-Are the results clearly and completely presented?

-Are the figures (Tables, Images) of sufficient quality for clarity?

Reviewer #2: yes

Reviewer #3: Results have not been significantly modified. Figure 3 included more information regarding geographical distribution of the species.

**Conclusions**

-Are the conclusions supported by the data presented?

-Are the limitations of analysis clearly described?

-Do the authors discuss how these data can be helpful to advance our understanding of the topic under study?

-Is public health relevance addressed?

Reviewer #2: yes

Reviewer #3: Data presented in the study does not disagree with the WHO Category 2 snakes' definition: "... the exact clinical data are still lacking and less frequently implicated". Although disability and mortality were not reported, a larger casuistics may be necessary to definitely suggest the exclusion of Tropidolaemus subannulatus of the Cat.2 list.

**Editorial and Data Presentation Modifications?**

Reviewer #2: high quality images are required.

Reviewer #3: Accept

**Summary and General Comments**

Reviewer #2: Authors have enhanced the MS.

I recommend its publication

Reviewer #3: (No Response)

PLOS authors have the option to publish the peer review history of their article (what does this mean?). If published, this will include your full peer review and any attached files.

Reviewer #2: No

Reviewer #3: No

---

## [Editor Report · Acceptance letter]

7 Dec 2022

Dear Dr Ismail,

We are delighted to inform you that your manuscript, "Frequency, clinical characteristics and outcomes of Tropidolaemus species bite envenomation in Malaysia," has been formally accepted for publication in PLOS Neglected Tropical Diseases.

Best regards,

Shaden Kamhawi

co-Editor-in-Chief

Paul Brindley

co-Editor-in-Chief
